# Indoor Bluetooth Low Energy Dataset for Localization, Tracking, Occupancy, and Social Interaction

**DOI:** 10.3390/s18124462

**Published:** 2018-12-17

**Authors:** Paolo Baronti, Paolo Barsocchi, Stefano Chessa, Fabio Mavilia, Filippo Palumbo

**Affiliations:** 1Institute of Information Science and Technologies, National Research Council, 56124 Pisa, Italy; paolo.baronti@cnr.it (P.B.); paolo.barsocchi@isti.cnr.it (P.B.); fabio.mavilia@isti.cnr.it (F.M.); 2Department of Computer Science, University of Pisa, 56127 Pisa, Italy; stefano.chessa@unipi.it

**Keywords:** indoor localization, tracking, social interaction, Bluetooth Low Energy, dataset

## Abstract

Indoor localization has become a mature research area, but further scientific developments are limited due to the lack of open datasets and corresponding frameworks suitable to compare and evaluate specialized localization solutions. Although several competitions provide datasets and environments for comparing different solutions, they hardly consider novel technologies such as Bluetooth Low Energy (BLE), which is gaining more and more importance in indoor localization due to its wide availability in personal and environmental devices and to its low costs and flexibility. This paper contributes to cover this gap by: (i) presenting a new indoor BLE dataset; (ii) reviewing several, meaningful use cases in different application scenarios; and (iii) discussing alternative uses of the dataset in the evaluation of different positioning and navigation applications, namely localization, tracking, occupancy and social interaction.

## 1. Introduction

The massive introduction of Internet of Things devices on the market—it is expected that a population of over 50 billions of devices will be connected to the Internet by 2020 [1,2]—and the consequent widespread availability of wearable cyber-physical devices opens new ways of delivering services to people. In this scenario, indoor localization plays a key role since it is a main enabler for location-based services that deliver services at the right time in the right place and that can thus better assist people in both their ordinary and extra-ordinary activities. At the same time, raw localization data have also been used to infer information about social interactions among people, and this use of localization data is gaining more and more attention, because of its potential use in online social networks (such as Facebook, Google+ or Twitter), mobile social networking [3] or mobile crowdsensing [4], where it can contribute to add a physical level of sociality in worlds that, up to now, had only been virtual. The need for wide-spread and low-cost solutions for indoor localization has pushed the research towards new technologies, solutions and algorithms. Currently, indoor localization is a live and active research area as witnessed by the large number of related publications and events. Since the very early period of research in the field, it became evident that the evaluation and comparison of different indoor localization solutions and technologies require experimentation in real environments with real equipment and users, with ensuing high costs for the setup and maintenance for the experimental testbeds. For this reason, researchers produce datasets [5,6] that can be freely used for experimentation, and several competitions on indoor localization [7,8] also offer the opportunity for researchers to test and compare their solutions, and to produce additional datasets. However, in most cases, such datasets focus on a specific measurement method and established technologies. In particular, datasets providing Received Signal Strength (RSS) data obtained with technologies such as WiFi and RFID are widely available [9], while only few datasets provide data obtained with more recent technologies such as Bluetooth Low Energy (BLE) [10]. Furthermore, many datasets are produced for the specific aim of evaluating one solution, or they are produced in an exploratory way during events (for example, [11,12]) but they often miss an accurate and publicly available ground truth. Finally, almost all datasets (and competitions too) address localization and (to a minor extent) tracking problems or room occupancy, but they hardly provide specific data aimed at detecting social interactions among users.

In this work, we propose a novel dataset that addresses all these limits. In particular, the dataset addresses BLE, since this is a low-cost technology available worldwide (all recent smartphones embed this technology), and RSS, since this is a simple and versatile way of measuring distances/proximity, which is also available to all wireless devices. The dataset was built with the purpose of providing an accurate ground truth and a large number of Received Signal Strength Indicator (RSSI) values obtained from both fixed and wearable BLE devices, so that is can be used to test solutions operating with different configurations (e.g., self/remote positioning and direct/indirect positioning) and a large number of use cases, including not only localization and tracking, but also room occupancy and social interactions among users. The dataset is freely accessible for research purposes without any limitation (the dataset can be freely downloaded from: http://wnlab.isti.cnr.it/localization).

The dataset was produced in a portion of our building that includes offices, corridors and public spaces by monitoring up to three human “actors” that act and move in these spaces. The installation includes a redundant number of fixed devices installed in each room or public space, and each actor carry two devices (in the hand and on the chest) to represent different kinds of use of wearable devices. Each device (either fixed or wearable) acts both as transmitter and receiver of the signal from any other transmitting device. Thus, the dataset can be used both in self positioning and in remote positioning scenarios. The actors move in the area by following several different scripts that model different use cases, from simple movements to meetings, to both test localization algorithms and detect social relationships among users. The ground truth is produced by the actors themselves that record their position over a large number of pre-defined signposts on the ground, by using a mobile application that we developed for the EvAAL localization competition [13,14]. The dataset we produced is composed by 2598 labeled points, collecting about 4 millions of RSSI values over 185 m^2^ walkable area. Finally, the paper presents a validation of the dataset conducted by assessing several data properties that are “neutral” with respect to the specific algorithms. Note that, although this approach does not focus on any specific algorithm (localization, proximity, tracking, detection of social interactions, etc.), it nevertheless implicitly provides an assessment of potential applications.

The rest of the paper is organized as follows: Section 2 reviews the state of the art in datasets, including both special-purpose datasets and exploratory datasets recorded at events or competitions. Section 3 presents the design of the data collection system and the methodology used in the production of the dataset and of the ground truth. Section 4 specifies the format of the dataset, including the specifications of the associated metadata, and Section 5 provides an insight on the dataset by showing relevant statistical analysis related to its coverage over the space and the correlation among different measurements over time and space. Section 6 draws the conclusions and the perspectives of the work.

## 2. Related Works

A lot of work has been conducted in the indoor positioning and navigation field by the research community. Outdoor positioning is already available with mostly good accuracy thanks to Global Navigation Satellite System (GNSS) receivers as de-facto standard. In indoor environments, where GNSS signals degrade too much to be reliable, other technologies have been explored for more than 20 years to get robust and accurate Indoor Positioning Systems (IPSs). Despite the good accuracy achieved by some systems, such as those based on sub-meter Ultra Wide Band (UWB) positioning [15,16], Ultrasound [17] and Visible Light [18], they still present barriers in their adoption due to costs for additional equipment and difficulties in deploying and maintaining dedicated infrastructure. Furthermore, there is the need to get reliable indoor positioning information with commodity devices, such as smart phone with embedded sensors and radio interfaces. For these reasons, the research on IPSs based on WiFi and Bluetooth RSS has been the most popular thanks to the ease of access of RSS information from basically any mobile device [19,20,21,22,23,24]. In this context, the need for common datasets or frameworks to compare and evaluate IPS solutions represents a big research challenge that has been faced by different works in our reference scenarios, with their strengths and shortcomings.

With regard to the localization and tracking scenario, the most exploited source of information is represented by RSS fingerprinting [12,25]. While for the WiFi technology there is plenty of datasets available, the same effort in terms of collected data is missing for BLE-based solutions. In [26], the authors investigated the impact of the number of WiFi fingerprinting points, number of samples, user orientation, and the issue of tracking a mobile user. In [27], the authors described a localization method based on a genetic algorithm presenting four experiments conducted in two different environments together with the collected data. The experiments presented in [28] are carried out at Tampere University of Technology (10,000 square meters approximately). The reference data are collected in 96 points and a mean of 30 RSSI measurements are computed. The experiments in [29] are performed in two buildings at the University of Minho, Portugal. A total of 392 calibration points are established in the whole environment and 9358 calibration samples are taken. The experiments done in [30] follow the comprehensive benchmarking methodology developed in the EVARILOS Project [31,32]. In [6], the authors presented an available database of RSSI values captured in the UJI University campus that covers a surface of 108,703 square meters, including three buildings with four or five floors, depending on the building.

In the occupancy detection scenario, many works have been developed addressing specific scenarios: surveillance [33]; dynamic allocation of space in commercial building [34]; Heating, Ventilation and Air Conditioning (HVAC) control [35]; and safety and security [36]. Among these scenarios, the estimation and detection of building occupancy related to efficient control of the energy consumption is the most studied. In [37], the authors presented a comprehensive review on building occupancy estimation and detection using different approaches based on the involved sensors. From the availability of data point of view, only environmental data have been presented as public datasets [38,39,40,41]. In these works, the occupancy detection is usually presented as a binary classification problem that requires the observation of environmental factors such as temperature and humidity to be used to classify whether a room is occupied or unoccupied. Very few works present results based on WiFi and BLE data [36,42,43] and no publicly available dataset is provided.

A different approach is usually adopted for the detection of social interactions. In this scenario, researchers exploit huge dataset of coarse-grained mobility traces to infer a high-level view of communities and trends in their common interests over the long period. In [44], the authors used three different datasets that capture human mobility over nine months based on online location-based social networks and cell phone location trace data. For location-based social networks, this means that a user checked-in to a specific location using the online social network website/application, and for cell phone data this means a user either initiated or received a phone call. In [45], the authors used data concerning social interaction and propinquity based on wireless and Bluetooth. The used dataset comprises experiments carried out considering four mobile devices carried by people sharing the same affiliation during their daily routines (commuting between home and office, going to leisure activities, and attending meetings in the office). Another approach is the one presented by Mtibaa et al. [46] using mobility traces form different datasets presenting implicit and explicit connections as ground truth. Implicit connections are defined as contacts between persons that share some common interests, while connections are defined as explicit only when the two nodes declared a direct connection (for example, links in applications such as Facebook). The authors used different datasets of traces with explicitly defined connections: MobiClique [47], composed of Bluetooth proximity data and RFCOMM data communications; and Infocom2006 [48], composed of Bluetooth sightings by groups of users carrying small devices (iMotes) for a number of days. The datasets of traces with implicit connections are Hope [49], composed of RFID tracking data collected from the seventh HOPE conference attendees carrying RFID badges, and a virtual mobility dataset composed of the traces of the virtual characters in the SecondLife virtual world [50]. A dataset providing fine-grained information, without using cameras [51,52], about the precise location of a socialization event in indoor environment is still missing.

To the extent of our knowledge, only few dataset are available based on BLE RSS that can be used for our reference scenarios. With regard to localization and tracking, Mohammadi et al. [53] presented a small dataset composed by 1046 labeled RSSI measurements from 13 BLE beacons collected in 106 still (from 10 to 20 s) positions. The dataset is gathered from a real-world deployment of a grid of iBeacons in a campus library area of 60 × 55 m. The authors did not describe the parameters used for transmitting the beacon advertisements, such as transmission power and frequency of transmission. In [54], the authors collected various measurement in an environment composed of six rooms (around 200 square meters in total) instrumented with custom BLE transmitters, performing 24 paths in total. The transmitters broadcast at 10 Hz but no transmission power is indicated. During the paths, a total of 125 positions are collected as waypoints. Particular attention to the socialization detection scenario is given by [55]. The dataset is composed of RSSI values collected from BLE beacons carried by people following their daily routine inside a university building for a whole month. A network of Raspberry Pi 3 (RPi)-based edge devices were deployed inside a multi-floor facility continuously gathering BLE advertisement packets and storing them in a cloud-based environment. Being the work focused on the mobile social network scenario, only mobile transmitters are considered, transmitting at 0 dBm and 1 Hz. The work involved 46 participants using 32 receivers RPi3 deployed on three 30 m × 60 m floors. An additional effect of being focused on this particular scenario is that no ground truth about location and occurrence of the socialization events is provided. Differently from the above datasets, our dataset addresses both localization and socialization data and it provides full information about the configuration of the devices and their deployment and the other relevant parameters of the data collection process. Furthermore, differently from the work in [55], we limit the number of people involved in the measurements but we provide a full ground truth about the position of each person.

## 3. Experimental Design and Methods

We designed the experimental campaign by considering two different configurations: BLE beacons transmitted by fixed stations (from now on called *anchors*) and received by mobile devices (namely, *self positioning configuration*); and BLE beacons transmitted by mobile tags worn by human actor(s) and received by the anchors (namely, *remote positioning configuration*).

### 3.1. The Environment

The test environment comprises a group of seven contiguous rooms located at our research facility, a connecting corridor and a small adjacent area housing coffee and vending machines. As can be seen in Figure 1, we laid out a grid of 277 points covering the walking surface of the test environment (desks and furniture were not covered by the grid). Each square in the figure is centered on a grid point and shows the associated numerical identifier. Squares with a cyan background represent seat positions that were used in the social interaction scenarios described later.

The environment was instrumented with 8 Raspberry Pi 3 devices, placed one in each of the rooms and in the coffee area in the positions indicated by the yellow circles in the figure. These fixed devices acted as *anchors* transmitting iBeacon advertisements at 10 Hz and with a transmission power varying from −18 dBm to 3 dBm (more details in Section 3.3).

We measured the coordinates of each point in cm with respect to the origin of our coordinate system, located at point 1 (at the extreme left of the corridor). X coordinates grow rightward while Y coordinates grow upward. Grid spacing was based on the floor tiles which are 60 cm squares. Some misalignment exists between tiles in the corridor/coffee area and those in the rooms. This is accurately reflected in the map. The total area covered amounts to about 185 m^2^ with a maximum horizontal span of ca. 16.6 m and maximum vertical span of ca. 14.3 m.

### 3.2. Reference Scenarios and Ground Truth Collection

The experimental campaign was performed with the aim of providing the most useful datasets for researchers investigating different fields of IPSs. We set up six reference scenarios (each corresponding to a produced dataset): *survey*, *indoor localization*, and four different *socialization* scenarios. Each scenario generated three different data collection campaigns, collecting RSSI values of iBeacon advertisements transmitted at three different powers: −18 dBm, −6 dBm, and +3 dBm. In total, we performed 18 data collection campaigns.

We started focusing on the fingerprinting-based methods for indoor positioning (this first run of experiments is collected in the dataset called *survey*). To this end, we performed a fine-grained survey of the area collecting RSSI values transmitted by the eight fixed beacons using a smart phone held in the hand of an actor at 1.2 m from the ground (*self positioning configuration*). In addition, the actor also wore a mobile beacon installed on a badge positioned on the actor’s chest (*remote positioning configuration*) and the relative RSSI values collected from each fixed receiver are also part of this dataset. Furthermore, to facilitate the developers of fingerprinting-based IPSs in considering different orientations of the device, the actor changed his orientation in each of the 277 positions every 2 s. Four different orientations were considered: with the device facing the [1 0 0] versor of the Cartesian plane representing the map (Orientation 1 in the dataset); facing the [0 −1 0] versor (Orientation 2); facing the [−1 0 0] versor (Orientation 3); and facing the [0 1 0] versor (Orientation 4).

To have a separate set of measurements to test fingerprinting-based IPSs, the same actor walked an additional path, as shown in Figure 2. RSSI values coming from the fixed beacons collected by the smart phone (*self positioning configuration*) and RSSI values coming from the worn mobile beacon collected by the fixed receivers (*remote positioning configuration*) were included in the dataset called *indoor localization*. The *indoor localization* path also included 13 points (indicated with numbers in Figure 2) in which the actor stood still for 10 s.

The experimental campaign for the social interaction scenarios was split in four sessions (thus, the *socialization* dataset comprises four parts), involving different numbers of actors. During the experimental campaign, the only persons inside the test area were the actors involved in the experiment. In particular, we selected two scenarios with two actors and two scenarios with three actors. In the latter cases, it is possible to evaluate how the NLOS and multipath, introduced by the third person, affect algorithms for the detection of social interactions. In the first social interaction scenario, Actor 1 moves at time t0 in order to meet Actor 2. At Time t1, the Meeting m1 (30 s of interaction) begins. At the end of the meeting (t2), Actor 1 goes back to his room (Figure 3a). In the second scenario (Figure 3b), the same two actors meet (Meeting m1) in the coffee area at Time t3 and walk together in the corridor until Time t4 and then go back to their respective offices. Scenarios 3 and 4 involved three actors. In the third scenario, three different meetings involving two actors were performed (Figure 3c). In particular, Meeting m1 is between Actors 1 and 2, Meeting m2 is between Actors 1 and 3, and finally Meeting m3 is between Actors 2 and 3. The first meeting takes place in the corridor at Time t3 between Actors 1 and 2. After 2 min Actor 1 moves in Room 1069 and the second meeting starts at Time t4. Meanwhile (Time t5), Actor 2 goes back to his office. Later, Meeting m2 ends and Actor 1 comes back in his office at Time t6. The last meeting starts at Time t8 between Actors 2 and 3. In the fourth scenario, different kinds of meetings involving two and three actors were performed. At first, a Meeting m1 between Actors 1 and 2 is performed at Time t3; Actor 3 then joins in generating a new Meeting m2 at Time t5. Afterwards, all actors walk together in the corridor and split up (Time t7) with Actor 1 going back to the office and the other two actors walking on together to the coffee area in the last Meeting m3 (Figure 3d). It should be noted that the last four social interaction scenarios can also be used as indoor localization datasets. In this case, the dataset provides insightful implications to be studied about possible interference introduced by other people moving in the environment represented by the three actors.

In total, we labeled 866 positions for each experimental campaign at three different transmission powers (3 dBm, −6 dBm, and −18 dBm) for a total of 2598 labeled points, obtaining almost three millions labeled RSSI values over 229 min of data collection campaign, with almost one hour of social interaction events. Table 1 provides additional details about the collected data. It should be noted that the overall duration and the total number of labeled positions are expressed as the sum of the three campaigns using different transmission powers (e.g., for the “Survey” scenario, each campaign lasted approximately 38 min collecting samples in each of the 277 points in the map shown in Figure 1). Furthermore, the amount of collected RSSI values is much less than what was transmitted due to the presence of packet loss, as explained in Section 5.

### 3.3. Sensing Infrastructure

The sensing infrastructure relied on BLED112 by Bluegiga (https://www.bluegiga.com/) as fixed transmitters and receivers, RadBeacon Dot produced by Radius Networks (http://store.radiusnetworks.com/collections/all/products/radbeacon-dot) as BLE tags and Honor 8 by Huawei Technologies (https://www.hihonor.com) as mobile receivers. BLED112 is a small USB dongle that integrates a 8051 microcontroller, a radio transceiver and a fully-compliant Bluetooth stack for application development. We used two dongles plugged into a Raspberry Pi, a very small single-board computer; the first one, set up as a receiver, was programmed to scan at a 100% duty cycle; the second one, as a transmitter, was programmed to send packets at 10 Hz with −18 dBm/−6 dBm/3 dBm of transmission power. Each Raspberry placed in the environment collected advertisements and, periodically, sent chunks of data to a web server via a WiFi/Ethernet connection. Analogously, an Android app, developed for Honor 8 smartphones, performed the same functionalities. The web server persisted data received from Raspberries and smartphones to a non-relational Mongodb database. RadBeacon Dots are really cheap and easy-to-configure devices, fully compliant with most BLE common implementation stacks, as iBeacon and Eddystone. Advertisement rate and transmission power can be tuned, ranging from 1 Hz to 10 Hz and from −18 dBm to 3 dBm, respectively. The dots were used as mobile transmitters and worn as badges by the actors. We performed three runs for each scenario defined in Section 3.2, corresponding to the transmission powers we have chosen, i.e., 3 dBm, −6 dBm, and −18 dBm, respectively.

## 4. Dataset Format

The dataset consists of different timeseries where each row represents a BLE signal collected by a receiver. The data format is shown in Table 2. All fields that make up the data format are numeric values. The entire dataset has been divided in two parts, one containing the self positioning data and one containing the remote positioning data.

Self and remote positioning sub-datasets have been further subdivided: each scenario of experimental campaign was replicated for each selected transmission power, i.e., 3 dBm, −6 dBm, and −18 dBm. Regarding ground truth annotations, the data format changes according to the reference scenario.

The fields composing the data format for the survey campaign are shown in Table 3. For each of the 277 positions labeled in Figure 1, we report the start time and the end time in which an actor stayed in that position, changing his orientation every 2 s. The “orientation” field, as described in Section 3.2, ranges from 1, when the actor faces the [1 0 0] versor of the Cartesian plane representing the map, to 4, when the actor faces the [0 1 0] versor, clockwise.

The data format of ground truth annotations for localization and social interaction scenarios is shown in Table 4. We report the timestamp when the actor passes through or reaches the target position.

## 5. Experimenting with the Dataset

We assessed the quality of the data gathering process (and thus of the dataset) by analyzing the packet loss, the RSS among devices, and the coverage for each scenario. In particular, we defined *packet loss rate* as the ratio between the number of beacons not received and the total number of beacons transmitted, and we evaluated the packet loss rate for each scenario and for each transmission power. We also defined *coverage* as the number of anchors seen by the mobile receiver (in the self positioning configuration), or as the number of anchors that receive the transmissions of the mobile device in the remote positioning configuration. We evaluated the coverage for each point of the survey scenario and for each transmission power. To show the effect of the multipath propagation in the collected dataset, we also analyzed the RSS between the an anchor and a mobile device. Moreover, we analyzed how the human presence affects the RSS for a specific couple of anchor and mobile device when changing the orientation of the actor. Finally, we analyzed the reciprocal RSSI values between two mobile devices (Mobiles 1 and 3) against their relative distance. We call this last property *channel symmetry*.

Note that, being an experimentation aimed at validating the dataset (and not algorithms), we focused these experiments on the assessment of the properties of the data themselves. Nevertheless, this evaluation also gives insights on the potential uses of the dataset in different applications. In particular, the coverage analysis implicitly relates to proximity and room occupancy, the analysis of RSS relates to fingerprinting algorithms for localization, while symmetry is closely related to algorithms for the detection of social interactions. The figures shown in this section further explain this approach for each of the analyzed metric.

The first step was to analyze the packet loss rate for all the scenarios by considering self and remote positioning. Table 5 shows the packet loss rate experimented by smart phones for all the three transmission powers used, i.e., 3 dBm, −6 dBm, and −18 dBm. As observed in the table, the loss rate ranges from 41% to 94%, depending on the receiving device and transmission power of the emitters: a high transmission power (3 dBm) determines a low loss rate, and vice versa. Loss rates of Mobile 1, handled by Actor 1, and Mobile 2, handled by Actor 2, are substantially comparable. Mobile 3 reveals a higher loss rate than the others: the reason can be found in the path followed by Actor 3 in Social Scenarios 3 and 4. Most of the time, Actor 3 remains in his position (114 in Figure 1), moving less in the environment than the other users and, therefore, collecting less beacons.

Table 6 and Table 7 report the packet loss rate experimented by the anchors. As expected, in this case, the loss rate experienced by the anchors that are located at the far end of the selected area (e.g., Anchors 1062 and 1080) is higher than what the anchors located in the central zone undergo. Please note that such a high packet loss is expected. It depends on the fact that the protocol does not provide QoS on overall end-to-end reliability (no acknowledgements and re-transmission of beacons) and on the channel (all communications are affected by obstacles such walls, furniture or people as well as by multipath). Furthermore, with low transmission power, more beacons are lost because a large number of receiving stations are out of range.

Figure 4 shows the coverage for each transmission power in the survey scenario. In particular, the heat maps show the number of anchors seen by Mobile 1 and the number of anchors that see Mobile 1 in self positioning and remote positioning scenarios, respectively. The heat map at 3 dBm in self positioning scenario underlines the occurred packet loss during the measurement campaign in six points. As shown in the figure, decreasing the transmission power also lowers coverage in both scenarios. Moreover, there is no significant difference in terms of coverage between the two scenarios. This result highlights the symmetric characteristics of the radio channel. When the transmission power is set at −18 dBm, a low value of coverage is measured, confirming that low transmission powers are suitable for proximity scenarios.

Figure 5 illustrates the RSSI between Mobile 1 and Anchor 1070 and vice versa, depending on the selected scenario. Radio reception at any given location depends on three main factors: free-space propagation loss, large-scale and small-scale fading [56]. Free-space propagation loss describes wave propagation without obstacles as a function of distance between transmitting and receiving antennas. Large-scale fading refers to shadowing by larger obstacles (such as buildings, with dimensions substantially larger than wavelength λ). Small-scale fading, in turn, is caused by multipath propagation and wave’s interference with its own delayed and attenuated reflections from surrounding objects. In the case of small-scale fading, areas of constructive and destructive interference are separated by half-wavelength distances, and received signal properties can rapidly change even after a small (sub-λ) receiver movement. For BLE signals, generated indoors by low-power transmitters, the main factors of spatial variability affecting the RSS are free-space loss and attenuation by walls and furniture. Due to the small wave-length (λ=12.5 cm), small-scale fading occurs at centimeter-scale distances, well below the calibration resolution of current BLE based positioning systems. As expected, in Figure 5, the RSSI values in each scenario more or less decrease when the distance between the receiver and the transmitter increases, due to walls and furniture.

Figure 6 gives a qualitative representation of the impact of the human orientation on the RSSI of packets exchanged between the Mobile 1 and the Anchor 1069 at 3 dBm. Indeed, when the mobile is turned in the direction of the anchor, the RSSI values are significantly higher than when the mobile is in the opposite direction. However, when the mobile is far from the anchor (i.e., the RSSI is below −90), there is no significant difference in terms of RSSI values between the two different orientations.

The aspects related to the channel symmetry between Devices 1 and 3 during the social interaction (Scenario 4) are shown in Figure 7, which also reports the relative distance between the two devices. When Actor 1 moves at Time t0, the mutual RSSI values decrease since the distance between the actors increases. When Actor 3 moves towards Actor 1 at Time t4, the RSSI values increase. During Meeting m2, although the distance is comparable with that at the beginning, the RSSI values are always greater. The figure highlights that, from the symmetry point of view, there is no significant difference when setting different transmission powers. However, when the transmission power is set to −18 dBm and the distance is greater than 4 m, the BLE packets are lost. It is important to note that, regardless of the transmission power chosen, during the meeting event (Meeting m2) between Actors 1 and 3 in Scenario 4, the RSSI values are always greater than what measured when there are no meeting events. Analyzing Figure 7, a potential use of the dataset to test algorithms for the detection of socialization events among users can also be deducted. Considering, for example, the case on the top of the figure, a simple algorithm that assumes that the two users are meeting by setting a threshold at −65 dBm for the RSSI detected at both receiver would be able, in most cases, to discriminate the cases in which the two users are far away from the cases in which the two users are close (and thus are possibly meeting each other). A similar test can be conducted in the other two configuration at different transmission powers (in the middle and in the bottom of the figure) by setting appropriate thresholds.

## 6. Conclusions

In the last years, in support to the increasing success of data-driven approaches in many fields of computer science and engineering, the production of datasets is becoming strategic. Indoor localization is not an exception and, in fact, several competitions and research groups have published indoor localization datasets resulting from their experiments and activities.

In this work, we make a step beyond, and we aim primarily at the production of a rich dataset, which is thus not just a byproduct, but it is the main product of this work. The dataset was produced by using a large, redundant number of devices (both fixed and wearable) that act at the same time in a scenario of self and remote positioning, where the users to localize are trained actors that move and meet according to a pre-defined script, designed to represent several use cases, including those of detection of social relationships among users. We show the flexibility of the dataset by analysing its properties and by exploring some relevant uses, concerning localization, proximity and detection of social relationships.

We expect the dataset will result useful to researchers and practitioners to experiment and test their solutions. In perspective, we also expect that more and more researchers will be engaged in the production of richer and general-purpose datasets, pushed by the need of data to feed new, data-driven algorithms concerning user mobility and sociality. Following this trend, our current and future work addresses the design of novel and more general datasets built using other localization technologies and in more varied scenarios.

## Figures and Tables

**Figure 1 sensors-18-04462-f001:**
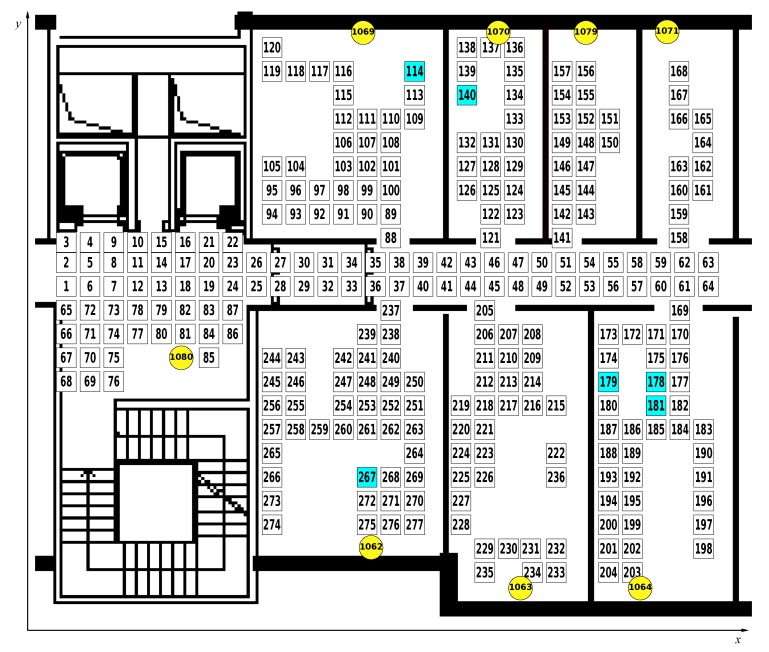
Grid layout for the test environment. Each point has X, Y coordinates.

**Figure 2 sensors-18-04462-f002:**
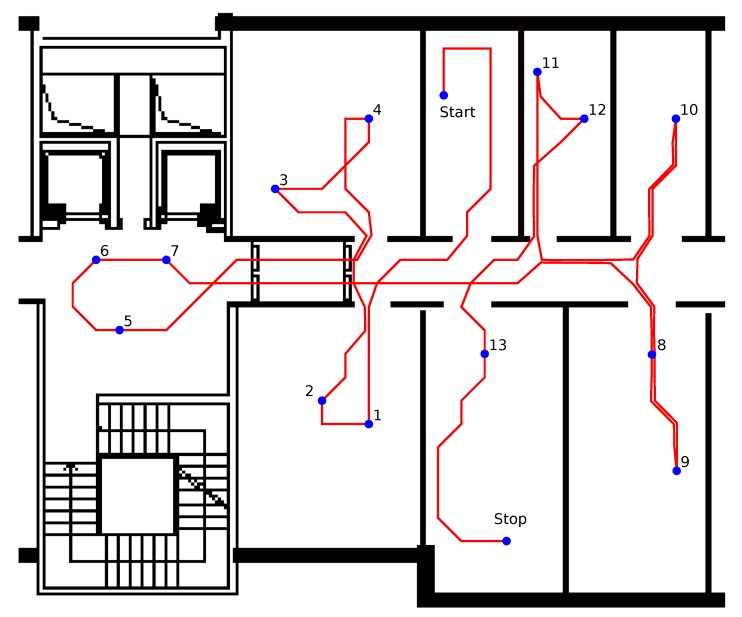
The path performed by the actor in the *indoor localization* scenario. The points indicated with numbers represent the positions in which the actor stood still for 10 s.

**Figure 3 sensors-18-04462-f003:**
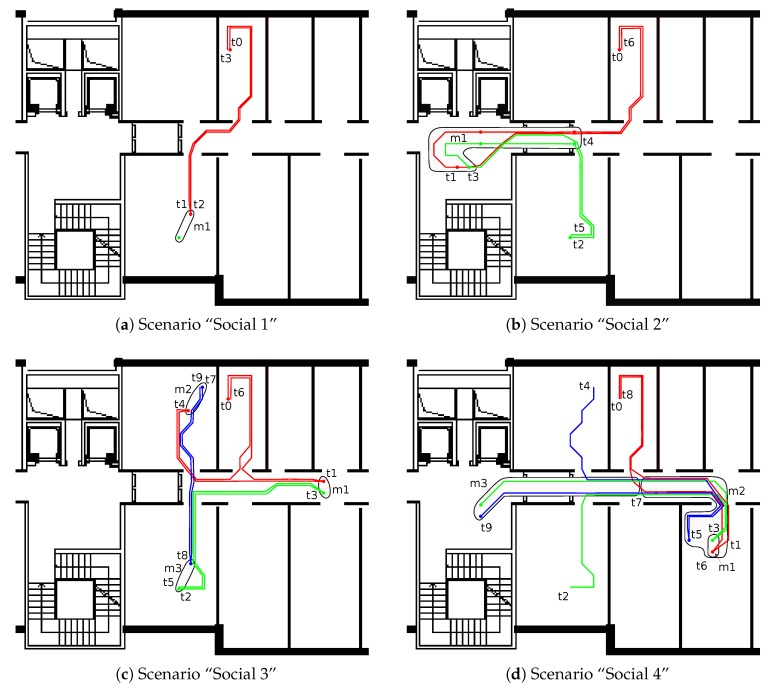
The four *social interaction* scenarios. Actors’ paths are indicated with different colors: red (Actor 1), green (Actor 2), and blue (Actor 3). Interactions indicated with gray areas $m_{i}$ at time $t_{i}$.

**Figure 4 sensors-18-04462-f004:**
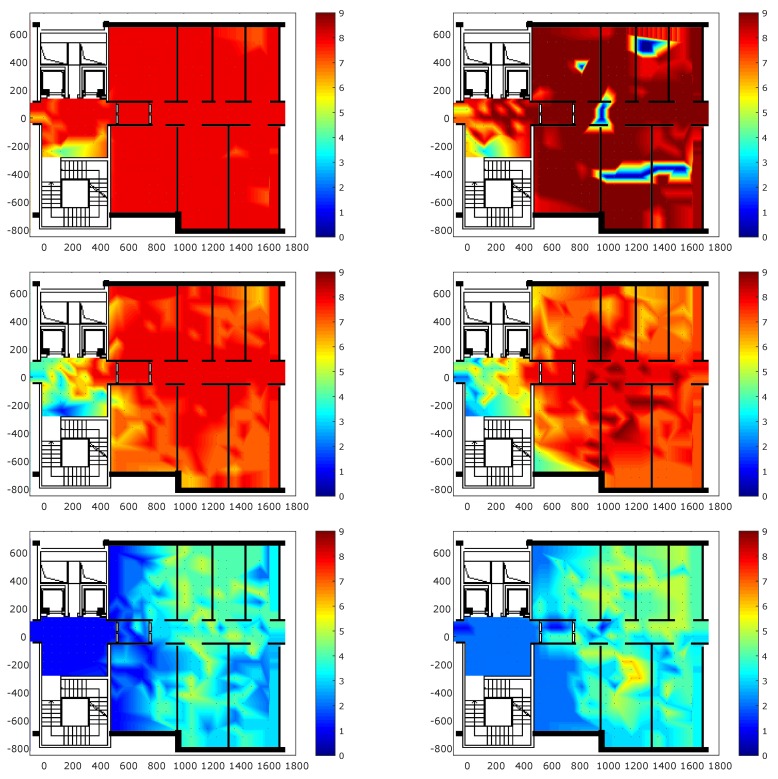
Coverage representation for the remote positioning (**left**) and self positioning configurations (**right**). From the top to bottom, the transmission power values are 3 dBm, −6 dBm, and −18 dBm.

**Figure 5 sensors-18-04462-f005:**
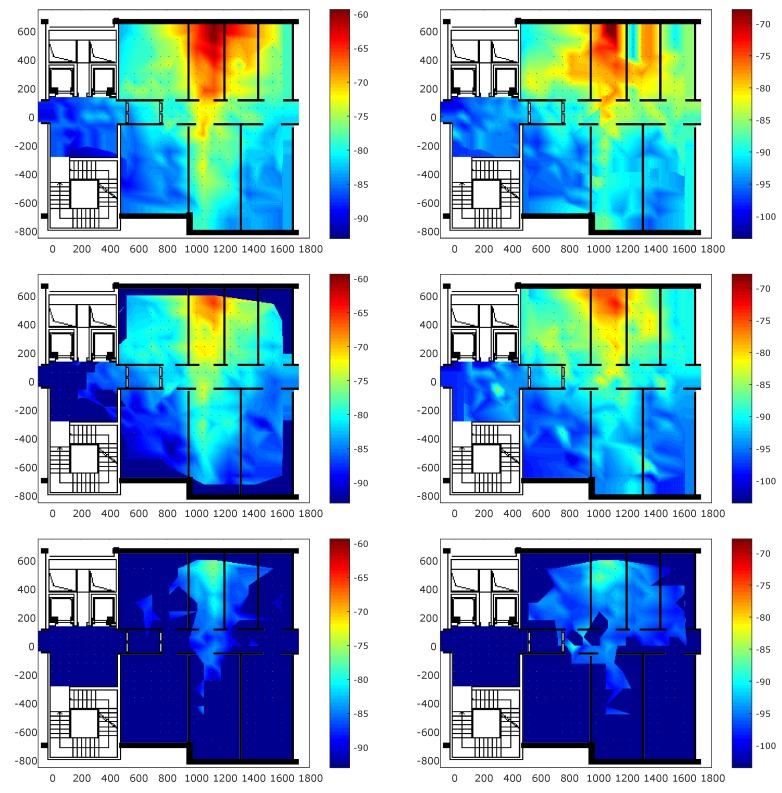
RSSI between Mobile 1 and the Anchor 1070 for both remote positioning (**left**) and self positioning configurations (**right**). From the top row to the bottom one, the transmission power values are 3 dBm, −6 dBm, and −18 dBm.

**Figure 6 sensors-18-04462-f006:**
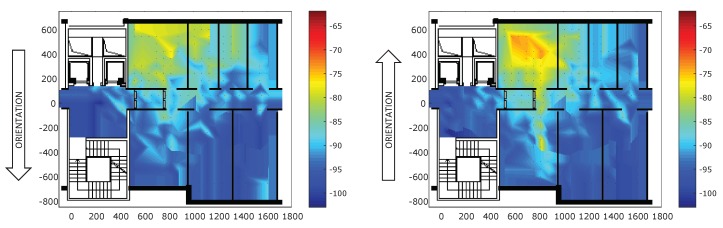
RSSI between Mobile 1 and Anchor 1069 at 3 dBm when the human orientation changes. On the left, the orientation corresponding to Position 2 in the dataset. On the right, the orientation corresponding to Position 4 in the dataset.

**Figure 7 sensors-18-04462-f007:**
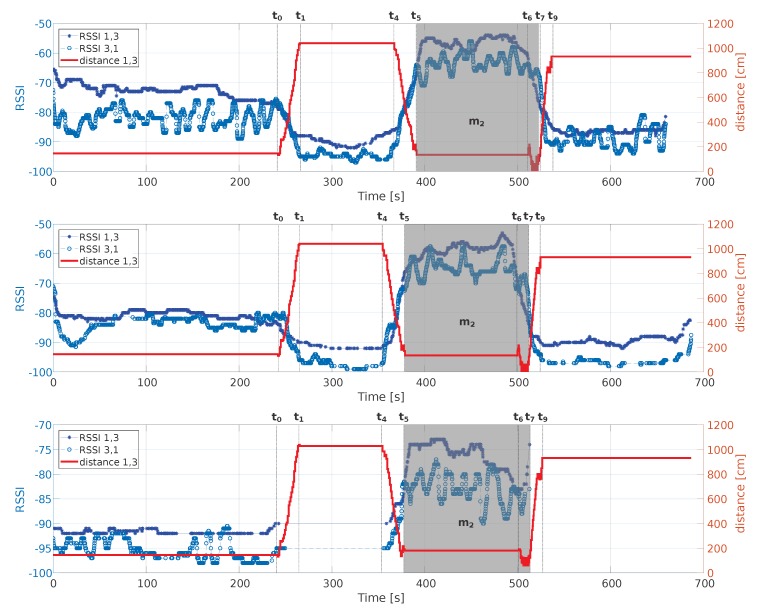
Symmetry analysis between Mobile 1 and Mobile 3 in accordance with the relative distance. Relevant timestamps related to the movements of Actors 1 and 3 in the scenario “Social 4” are shown, together with the social interaction period Meeting m2 as grey area. From the top to bottom, the transmission power values are: 3 dBm (**top**); −6 dBm (**middle**); and −18 dBm (**bottom**).

**Table 1 sensors-18-04462-t001:** Dataset overview. The number of labeled positions during a meeting and the total duration of meetings are shown in brackets.

	Labeled Positions			Collected RSSI Values						Duration [min]
	Actor 1	Actor 2	Actor 3	3 dBm		−6 dBm		−18 dBm		
				Remote	Self	Remote	Self	Remote	Self	
Survey	831	-	-	578,173	99,050	302,649	72,523	24,006	31,187	114.6
Localization	453	-	-	80,185	15,862	40,845	10,267	3923	4459	15
Social 1	135 (3)	3 (3)	-	80,327	29,346	42,476	26,947	4196	11,462	14.1 (9)
Social 2	180 (45)	132 (42)	-	86,989	34,754	48,417	26,112	4436	13,691	15.9 (6.6)
Social 3	183 (6)	129 (6)	93 (6)	250,546	103,786	146,653	78,460	14,401	28,161	36 (18)
Social 4	150 (39)	153 (78)	156 (81)	215,643	80,075	137,388	54,384	16,030	29,232	33 (18)
**Total**	**1932** (**93**)	**417** (**129**)	**249** (**87**)	**1291,863**	**362,873**	**718,428**	**268,693**	**66,992**	**118,192**	**229** (**51.6**)

**Table 2 sensors-18-04462-t002:** Data format of each BLE signal collected by receivers.

EpochTime	Receiver ID	Sender ID	RSSI

**Table 3 sensors-18-04462-t003:** Data format of ground truth annotations for the survey scenario.

Start EpochTime	End EpochTime	Actor ID	Position	Orientation

**Table 4 sensors-18-04462-t004:** Data format of ground truth annotations for localization and social interaction scenarios.

EpochTime	Actor ID	Position

**Table 5 sensors-18-04462-t005:** Packet loss rate experimented by smart phones.

	Mobile 1			Mobile 2			Mobile 3		
Scenario	3 dBm	−46 dBm	−18 dBm	3 dBm	−6 dBm	−18 dBm	3 dBm	−6 dBm	−18 dBm
Survey	52.10	64.74	84.92	-	-	-	-	-	-
Localization	41.64	62.22	83.59	-	-	-	-	-	-
Social 1	45.36	59.09	84.67	51.67	46.35	74.82	-	-	-
Social 2	55.05	67.12	78.78	36.00	51.02	78.29	-	-	-
Social 3	45.55	59.89	80.06	40.58	55.29	86.39	82.09	86.15	94.75
Social 4	41.95	57.74	79.59	63.10	82.26	89.03	84.48	88.01	94.57

**Table 6 sensors-18-04462-t006:** Packet loss rate experimented by Anchors 1062, 1063, 1064 and 1069.

	Station 1062			Station 1063			Station 1064			Station 1069		
Scenario	3 dBm	−6 dBm	−18 dBm	3 dBm	−6 dBm	−18 dBm	3 dBm	−6 dBm	−18 dBm	3 dBm	−6 dBm	−18 dBm
Survey	66.40	93.43	98.99	54.64	73.62	99.31	55.93	77.26	97.49	63.40	81.53	99.08
Localization	74.35	92.34	99.16	48.23	72.24	99.20	48.73	81.87	98.05	50.27	76.37	99.25
Social 1	64.91	82.74	95.23	53.60	76.01	99.81	44.89	70.95	99.04	52.20	77.73	99.11
Social 2	71.51	89.80	99.52	61.39	77.37	99.93	55.74	75.15	99.02	54.20	77.08	99.58
Social 3	64.29	81.90	98.35	54.00	74.36	99.81	46.07	72.82	99.05	50.94	71.51	92.40
Social 4	63.82	86.15	99.83	54.13	72.46	99.74	48.38	69.86	97.62	54.04	74.60	95.06

**Table 7 sensors-18-04462-t007:** Packet loss rate experimented by Anchors 1070, 1071, 1079 and 1080.

	Station 1070			Station 1071			Station 1079			Station 1080		
Scenario	3 dBm	−6 dBm	−18 dBm	3 dBm	−6 dBm	−18 dBm	3 dBm	−6 dBm	−18 dBm	3 dBm	−6 dBm	−18 dBm
Survey	50.96	72.41	98.71	56.46	70.05	95.63	47.26	70.18	98.92	90.42	95.98	98.82
Localization	49.68	72.04	96.68	51.17	69.31	95.65	56.04	70.12	96.81	89.61	96.60	98.94
Social 1	61.87	77.08	95.32	59.88	75.24	96.79	59.50	76.33	98.20	89.95	98.29	99.99
Social 2	57.92	74.62	95.44	57.40	73.85	96.41	58.18	75.44	98.23	80.64	88.01	96.38
Social 3	51.02	67.30	94.10	49.69	67.36	96.83	49.72	65.74	97.64	84.30	96.14	99.97
Social 4	51.95	72.31	94.55	63.77	67.93	96.04	51.32	66.42	98.15	85.33	90.25	96.77

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
