# Peer review of "Indoor Bluetooth Low Energy Dataset for Localization, Tracking, Occupancy, and Social Interaction"

_sensors, 2018, doi:10.3390/s18124462_

Reviewer 1 Report

The paper is generally well motivated, structured, written and referenced. I think that the dataset made available will be useful for the community. The literature review and the references are also particularly useful and rather complete and updated. Apart of a few typos (that would be corrected in the editor process afterwards) my only comments to some aspects that still could be improved in a final writing are:

1. Figure 1: Monitor (RPi3) #1000 should read #1080

2. Text description of Figure 3 (text on page 6) could be more clear describing the social interactions. Also, in that figure, the reader could be a little bit confused about the exact values of timestamps t1, t2, ..., to relate them with posterior figures (such as Fig. 7, for example). 

3. In this same vein, this last figure 7 could be labelled with this additional information, just for the sake of clearness. It is not completey clear to me, for example, why the red line appears with apparent sampling discontinuities, and certainly it would be useful to be able to relate the time line with the social interactions shown in figure 3, for example.

4. Given that I found the literature review rather interesting and complete, a pair of papers that are certainly very related to this work were recently presented in Mobiquitous 2018 (http://mobiquitous.org/accepted-papers/). I give the references here which, though are still in press, could be useful for the authors if they wish to cite them:

- "bTracked: Highly Accurate Field Deployable Real-Time Indoor Spatial Tracking for Human Behavior Observations", M. Chesser, L. Chea, H. Van Nguyen, D.C. Ranasinghe. MobiQuitous ’18, November 5–7, 2018, New York,USA. ACM Digital Library, DOI: 10.1145/3286978.3286986

- "Beyond the RSSI value in BLE-based passive indoor localization: let data speak", P.E. López-de-Teruel, O. Canovas, F.J. García, R. Gonzalez, J.A. Carrasco. MobiQuitous ’18, November 5–7, 2018, New York,USA. ACM Digital Library, DOI: 10.1145/3286978.3287020

Author Response

Response to Reviewer 1 Comments

Comments and Suggestions for Authors

The paper is generally well motivated, structured, written and referenced. I think that the dataset made available will be useful for the community. The literature review and the references are also particularly useful and rather complete and updated. Apart of a few typos (that would be corrected in the editor process afterwards) my only comments to some aspects that still could be improved in a final writing are:

Point 1: Figure 1: Monitor (RPi3) #1000 should read #1080.

Response 1: We thank the reviewer for pointing out the typo. It has been corrected in the figure.

Point 2: Text description of Figure 3 (text on page 6) could be more clear describing the social interactions. Also, in that figure, the reader could be a little bit confused about the exact values of timestamps t1, t2, ..., to relate them with posterior figures (such as Fig. 7, for example).

Response 2: We better described the social interaction in Section 3.2 by highlighting what t1, t2, … means. In particular, we partially rewrote (lines from 205 to 228) the description of the social interaction campaign and we modified Figure 7 in order to represent the timestamp previously reported in Figure 3d.

Point 3: In this same vein, this last figure 7 could be labelled with this additional information, just for the sake of clearness. It is not completely clear to me, for example, why the red line appears with apparent sampling discontinuities, and certainly it would be useful to be able to relate the time line with the social interactions shown in figure 3, for example.

Response 3: We modified Figure 7 adding the timestamp labels as proposed by the reviewer and we modified the description of the figure accordingly to reflect the changes (lines from 339 to 349). In the Figure, each time an actor moves, the distance between the two actors is computed. Therefore, the sampling discontinuities of the red line are produced when all the actors remain in the same position over time.

For the sake of clearness, we completely modified Figure 7, drawing the red line without star dots and adding information about the social interactions also in the caption.

Point 4: Given that I found the literature review rather interesting and complete, a pair of papers that are certainly very related to this work were recently presented in Mobiquitous 2018 (http://mobiquitous.org/accepted-papers/). I give the references here which, though are still in press, could be useful for the authors if they wish to cite them:

- "bTracked: Highly Accurate Field Deployable Real-Time Indoor Spatial Tracking for Human Behavior Observations", M. Chesser, L. Chea, H. Van Nguyen, D.C. Ranasinghe. MobiQuitous ’18, November 5–7, 2018, New York,USA. ACM Digital Library, DOI: 10.1145/3286978.3286986

- "Beyond the RSSI value in BLE-based passive indoor localization: let data speak", P.E. López-de-Teruel, O. Canovas, F.J. García, R. Gonzalez, J.A. Carrasco. MobiQuitous ’18, November 5–7, 2018, New York,USA. ACM Digital Library, DOI: 10.1145/3286978.3287020

Response 4: We thank the reviewer for the suggestion. We found the papers very interesting and perfectly fitting the scope of our paper. We added references [22] and [23] in Section 2 - Related Works.

Reviewer 2 Report

The authors present an interesting paper about Indoor localization based on Datasets aquired from BLE devices. However they must improve some issues, namely:

- The characterization of the tests should be better described. For example, the values in Table 1, in particular the ratio of the samples versus time, are difficult to understand. For "survey" gives on average 4s / sample, for "Localization" and "Social1", gives on average 2s / sample. For other cases it is not easy to find a relationship.Did the authors not previously report that every 2s were performing a sample?

The error rate is too high, with no clear explanations for this in the text. Did this occur during the site survey and during the tests?

In the situations of characterization of "social events", the authors refer 3 actors to perform tests, what was the occupation density of the events? this is important to clarify because there may be situations of NLOS, and the effect of the body, right?

Leaving such an interesting article to the mere construction of a "Data set" seems little. Have the authors not performed some validation tests? Just wanted to create a data set?

In the abstract, do they refer to "discussion of alternative uses" about the data set created to evaluate different localization and navigation algorithms? Have the authors performed some tests for data set validation?

Congratulation on your work.

Author Response

Response to Reviewer 2 Comments

Comments and Suggestions for Authors

The authors present an interesting paper about Indoor localization based on Datasets acquired from BLE devices. However they must improve some issues, namely:

Point 1: The characterization of the tests should be better described. For example, the values in Table 1, in particular the ratio of the samples versus time, are difficult to understand. For "survey" gives on average 4s / sample, for "Localization" and "Social1", gives on average 2s / sample. For other cases it is not easy to find a relationship. Did the authors not previously report that every 2s were performing a sample?

Response 1: We thank the reviewer for pointing out an actual lack of clarity in the description of the tests. We improved the description of the tests conducted (lines from 205 to 228) and the values in Table 1 (please also refer to response 2 and 3 to reviewer 1).

In particular, Table 1 previously reported all the collected data also between two consecutive labelled point creating a possible discrepancy with the total durations for each scenario. Considering the reviewer’s comment and after an internal review, we preferred to show only the amount of data actually labelled as quantitative metric for the dataset. This also clarifies the numbers of sample per seconds collected. Anyway, please note that the number of transmitted beacons does not reflect the number of beacons collected, due to inherent nature of the BLE advertisement protocol (Point 2). Tables 5, 6, and 7 have also been modified accordingly.

Furthermore, the indicated total duration for each scenario is the sum of the durations of the three campaigns at different transmission powers. In the case of “Survey” in the remote positioning configuration, the campaign lasted ~38 minutes at each transmission power obtaining: 38 * 60 (seconds) * 10 (beacons transmitted each seconds) * 8 (number of transmitters) * 8 (number of receivers) = 1459200 expected collected beacons. Considering an average packet loss of 0.6 at a TX power of 3dBm, we obtain around 580000 collected beacons, in line with what is now shown in Table 1. The same computation applies to the other scenarios and configurations.

For the sake of clarity, we also added these observations in the paper (lines from 232 to 237):

“Table 1 provides additional details about the collected data. It should be noted that the overall duration and the total number of labelled positions are expressed as the sum of the three campaigns using different transmission powers (e.g., for the ``Survey'' scenario, each campaign lasted approximately 38 minutes collecting samples in each of the 277 points in the map shown in Figure 1). Furthermore, the amount of collected RSSI values is much less than what transmitted due to the presence of packet loss as will be explained in Section 5.”

Point 2: The error rate is too high, with no clear explanations for this in the text. Did this occur during the site survey and during the tests?

Response 2: We understand that the reviewer refers to the high packet loss here. Actually that is very high, but it is expected, and should be like this. In our case, in fact, packet loss depends on the protocol (which does not provide QoS for the beacons) and on the channel (since presence of obstacles like walls and people and multipath effects affect very much the indoor communications). Furthermore, we collected data testing different transmission power, some of them very low (very low transmission power is useful for proximity detection for example), which implies that many beacons cannot be received by the receivers out of range (for example those placed in different rooms).

We clarified this in Section 5 (lines from 304 to 309) as follows:

“Please note that such a high packet loss is expected. It depends on the fact that the protocol does not provide QoS on overall end-to-end reliability (no acknowledgements and retransmission of beacons) and on the channel (all communications are affected by obstacles like walls, furniture or people and also by multipath). Furthermore, with low transmission power, more beacons are lost because a large number of receiving stations are out of range.”

Point 3: In the situations of characterization of "social events", the authors refer 3 actors to perform tests, what was the occupation density of the events? This is important to clarify because there may be situations of NLOS, and the effect of the body, right?

Response 3: The persons inside the area of the test were only the actors involved in the measurement campaigns. We modified the description of the experimental campaign for the social interaction in order to better clarify this point (lines from 205 to 207).

We agree with the reviewer on the importance of NLOS and multipath effects of external persons. That is why we considered two scenarios with only 2 actors and two specific scenarios with 3 actors. In this case, it is possible to evaluate how the NLOS and multipath affect algorithms for the detection of social interactions (lines from 208 to 210).

Point 4: Leaving such an interesting article to the mere construction of a "Data set" seems little. Have the authors not performed some validation tests? Just wanted to create a data set?

In the abstract, do they refer to "discussion of alternative uses" about the data set created to evaluate different localization and navigation algorithms? Have the authors performed some tests for data set validation?

Response 4: We really thought very much about the purpose of this work and about its motivations since its first planning, and we understand the concern of the reviewer, because it was also our concern. Our purpose is, in fact, to go beyond a mere construction of a dataset, by defining a methodology both for its construction and for its validation. We were moved by a need we encountered several time in our work, since many existing datasets (even public) are produced for the evaluation of a specific algorithm or solution, while we believe this field to be already mature to start producing “neutral” datasets that, as observed by Reviewer 3, “will promote the development of these areas”.

Being this the purpose, we realized that a “conventional” validation obtained by experimenting the dataset with some sample algorithms was not what we needed, because it does not allow to distinguish the impact on the results of the validation due to the algorithm from those due to the data themselves (in fact, this is usually done to validate an algorithm). For this reason, our validation strategy was entirely focused on the data themselves, by analysing properties that are “neutral”, with respect to the different algorithms. Although this approach does not focus on any specific algorithm (either of localization, proximity, tracking, detection of social interactions, etc...), it nevertheless implicitly provides an assessment of potential applications. For instance, the heatmaps in Figures 5 and 6 are directly related to fingerprinting for localization; the heatmaps in Figure 4 can be related to proximity and room occupancy, especially those obtained with lower transmission power; while Figure 7 implicitly refers to algorithms for the detection of social interactions.

We realize now, on the base of the comment of the reviewer, that this validation strategy (along with its motivation and its implications on the potential applications) was not explained at all in the paper. We have thus added the following text in the second paragraph in Section 5 (lines from 285 to 291) to fill this gap:

“Note that, being an experimentation aimed at validating the dataset (and not algorithms), we focused these experiments on the assessment of the properties of the data themselves. Nevertheless, this evaluation also gives insights on the potential uses of the dataset in different applications. In particular, the coverage analysis implicitly relates to proximity and room occupancy, the analysis of RSS relates to fingerprinting algorithms for localization, while symmetry is closely related to algorithms for the detection of social interactions. The figures shown in this section further explain this approach for each of the analyzed metric.”

And we also have added this statement in the introduction (lines 63 to 66):

“Finally, the paper presents a validation of the dataset conducted by assessing several properties of the data themselves that are “neutral” with respect to the specific algorithms. Note that, although this approach does not focus on any specific algorithm (either of localization, proximity, tracking, detection of social interactions, etc.), it nevertheless implicitly provides an assessment of potential applications.”

Reviewer 3 Report

This paper presents a dataset for localization, tracking, occupancy, and social interaction. A public dataset for fair comparison on different algorithms will promote the development of these areas. The paper is well-organized and well-written. It will be good if this work can take these minor concerns into consideration, which are shown as follows:

1.       The review on occupancy detection is too brief (page 3, line 99-103), which is awkward. Since occupancy detection is also importance, the content should be balanced with other topics. A new survey paper (Building occupancy estimation and detection: A review (2018)) may give you some hint.

2.       The analysis on packet loss, coverage and symmetry is impressive. However, another important indicator for a good dataset is the evaluation results. It is suggested to run some simple benchmark algorithms on this dataset to show the feasibility of the proposed dataset and encourage other researchers to perform comparison on this dataset.

Before that, I also suggest the authors to include more related works on this field, I list some as follows:

“Location fingerprinting with bluetooth low energy beacons”

“Improving indoor localization using bluetooth low energy beacons”

“Smartphone inertial sensor-based indoor localization and tracking with iBeacon corrections”

Author Response

Response to Reviewer 3 Comments

Comments and Suggestions for Authors

This paper presents a dataset for localization, tracking, occupancy, and social interaction. A public dataset for fair comparison on different algorithms will promote the development of these areas. The paper is well-organized and well-written. It will be good if this work can take these minor concerns into consideration, which are shown as follows:

Point 1: The review on occupancy detection is too brief (page 3, line 99-103), which is awkward. Since occupancy detection is also importance, the content should be balanced with other topics. A new survey paper (Building occupancy estimation and detection: A review (2018)) may give you some hint.

Response 1: We agree with the reviewer that the analysis of state of the art in the occupancy detection was missing the right emphasis. We thank the reviewer for the suggestion on the recent survey. We took inspiration from it for expanding our review, adding more interesting papers in related scenarios. Indeed, the following text was added in Section 2 (lines from 104 to 114):

“In the occupancy detection scenario, a large number of works have been developed addressing specific scenarios: surveillance [33]; dynamic allocation of space in commercial building [34]; Heating, Ventilation and Air Conditioning (HVAC) control [35]; safety and security [36]. Among these scenarios, the estimation and detection of building occupancy related to efficient control of the energy consumption is the most studied. In [37], authors present a comprehensive review on building occupancy estimation and detection using different approaches based on the involved sensors.

From the availability of data point of view, only environmental data has been presented as public datasets [38-41]. In these works, the occupancy detection is usually presented as a binary classification problem which requires the observation of environmental factors such as temperature and humidity to be used to classify whether a room is occupied or unoccupied. Very few works present results based on WiFi and BLE data [36,42,43] and no publicly available dataset is provided.”

Point 2: The analysis on packet loss, coverage and symmetry is impressive. However, another important indicator for a good dataset is the evaluation results. It is suggested to run some simple benchmark algorithms on this dataset to show the feasibility of the proposed dataset and encourage other researchers to perform comparison on this dataset.

Response 2: We thank the reviewer for describing the analysis as “impressive”. Our intention, in fact, was to produce a “neutral” dataset, independent of the potential solutions to test on it, and for this we devised a validation strategy that is entirely focused on the data rather than on the algorithms. This because validating the data with the application of algorithms would have impaired the validation of the data themselves… Furthermore, it would have not allowed to distinguish the impact on the results of the validation due to the algorithm from those due to the data themselves. However, although our approach does not focus on any specific algorithm (either of localization, proximity, tracking, detection of social interactions, etc.), it nevertheless implicitly provides an assessment of potential applications. For instance, the heatmaps in Figures 5 and 6 are directly related to fingerprinting for localization; the heatmaps in Figure 4 can be related to proximity and room occupancy, especially those obtained with lower transmission power; while Figure 7 implicitly validates even simple algorithms for the detection of social interactions.  Since an explanation of our validation strategy and of its motivation was missing, we added the following statement in the second paragraph of Section 5 (lines from 285 to 291) to state it:

“Note that, being an experimentation aimed at validating the dataset (and not algorithms), we focused these experiments on the assessment of the properties of the data themselves. Nevertheless, this evaluation also gives insights on the potential uses of the dataset in different applications. In particular, the coverage analysis implicitly relates to proximity and room occupancy, the analysis of RSS relates to fingerprinting algorithms for localization, while symmetry is closely related to algorithms for the detection of social interactions. The figures shown in this section further explain this approach for each of the analyzed metric.”

For what concerns the presentation of a use case, considering that the use of state of the art algorithms would bring us out of the scope of the paper, we decided instead to present in Section 5 a simple example of how the dataset could be used. To this purpose we chose the case of detection of socialization events which is the “less common” application. In particular, we have added the following statement at the end of Section 5 (lines from 349 to 356):

“Analyzing Figure 7, it can also be deducted a potential use of the dataset to test algorithms for the detection of socialization events among users. Considering, for example, the case on the top of the figure, a simple algorithm that assumes that the two users are meeting by setting a threshold at -65 dBm for the RSSI detected at both receiver would be able, in most cases, to discriminate the cases in which the two users are far away from the cases in which the two users are close (and thus are possibly meeting each other). A similar test can be conducted in the other two configuration at different transmission powers (in the middle and in the bottom of the figure) by setting appropriate thresholds.”

Point 3: Before that, I also suggest the authors to include more related works on this field, I list some as follows:

“Location fingerprinting with bluetooth low energy beacons”

“Improving indoor localization using bluetooth low energy beacons”

“Smartphone inertial sensor-based indoor localization and tracking with iBeacon corrections”

Response 3: We thank the reviewer for the suggestion. We found the papers very interesting and perfectly fitting our scope. We added references [12], [24], and [25], together with brief comments on them, in Section 2 - Related Works.

Round  2

Reviewer 3 Report

Thank you for addressing this reviewer's concerns.